# Morphometric Study of the Initial Ventricular Indices to Predict the Complications and Outcome of Aneurysmal Subarachnoid Hemorrhage

**DOI:** 10.3390/jcm12072585

**Published:** 2023-03-29

**Authors:** Maryam Said, Meltem Gümüs, Jan Rodemerk, Mehdi Chihi, Laurèl Rauschenbach, Thiemo F. Dinger, Marvin Darkwah Oppong, Yahya Ahmadipour, Philipp Dammann, Karsten H. Wrede, Ulrich Sure, Ramazan Jabbarli

**Affiliations:** 1Department of Neurosurgery and Spine Surgery, University Hospital of Essen, 45147 Essen, Germany; 2Center for Translational Neuro & Behavioral Sciences (C-TNBS), University Duisburg Essen, 45127 Essen, Germany

**Keywords:** subarachnoid hemorrhage, ventricular measurements, inflammation, marker, decompressive craniectomy

## Abstract

Objective: Acute hydrocephalus is a common complication in patients with aneurysmal subarachnoid hemorrhage (SAH). Several ventricular indices have been introduced to enable measurements of ventricular morphology. Previously, researchers have showed their diagnostic value for various neurological disorders. In this study, we evaluated the association between ventricular indices and the clinical course, occurrence of complications and outcome of SAH. Methods: A total of 745 SAH patients with available early admission computed tomography scans were included in the analyses. Six ventricular indices (bifrontal, bicaudate, ventricular and third ventricle ratios and Evans’ and Huckman’s indices) were measured. Primary endpoints included the occurrence of cerebral infarctions, in-hospital mortality and a poor outcome at 6 months. Secondary endpoints included different adverse events in the course of SAH. Clinically relevant cut-offs for the indices were determined using receiver operating curves. Univariate analyses were performed. Multivariate analyses were conducted on significant findings in a stepwise backward regression model. Results: The higher the values of the ventricular indices were and the older the patient was, the higher the WFNS and Fisher’s scores were, and the lower the SEBES score was at admission. Patients with larger ventricles showed a shorter duration of intracranial pressure increase > 20 mmHg and required decompressive craniectomy less frequently. Ventricular indices were independently associated with the parameters of inflammatory response after SAH (C-reactive protein in serum and interleukin-6 in cerebrospinal fluid and fever). Finally, there were independent correlations between larger ventricles and all the primary endpoints. Conclusions: The lower risk of intracranial pressure increase and absence of an association with vasospasm or systemic infections during SAH, and the poorer outcome in individuals with larger ventricles might be related to a more pronounced neuroinflammatory response after aneurysmal bleeding. These observations might be helpful in the development of specific medical and surgical treatment strategies for SAH patients depending on the initial ventricle measurements.

## 1. Introduction

Subarachnoid hemorrhage (SAH) after an intracranial aneurysm rupture is known to have several severe complications. Acute hydrocephalus in SAH is a very frequent occurring phenomenon, with reported rates of up to 97% [1]. Ventricular morphology, and especially its pathological enlargement, have been a subject of interest for a long time. Anatomical studies on the brain have been conducted since ancient times, and in around the 1500 s, Leonardo da Vinci managed to make a cast out of the ventricular system of an ox [2]. Over time, several animal and human cadaver studies have been conducted on the brain with descriptions of ventricular morphology [3]. The introduction of radiographic imaging immensely aided research on the ventricular system. Ventricular enlargement on a pneumocephalogram was described by Evans in 1942 [4]. The simple index measurement he presented, consisting of the ratio of the maximal diameter of the frontal horns to the internal diameter of the cranial vault, is currently still in use. Since then, several other ventricular measurements have been introduced. Correlations between these measurements and different diseases, such as normal pressure hydrocephalus [5], Morbus Alzheimer [6] and schizophrenia [7], have been described. Ventricular measurements have been incorporated in the guideline for normal pressure hydrocephalus as a diagnostic tool [5]. This illustrates the usefulness ventricular measurements can provide in clinical practice. With the frequent occurrence of acute hydrocephalus in SAH, ventricular measurements may contribute to our advanced understanding of this complicated disease. Previously, it has been shown that in SAH patients with acute hydrocephalus, the cerebral blood flow is reduced, especially in the periventricular regions [8]. Additionally, a correlation between a poor functional outcome after SAH and acute hydrocephalus has been reported [9]. 

Quantifying this acute hydrocephalus could aid in the prediction of other SAH complications and the final outcome. The aim of our study was to evaluate the association between ventricular indices and the clinical course, occurrence of complications and outcome of SAH by using morphometric data of several ventricular indices. 

## 2. Methods

### 2.1. Patient Population

For our retrospective cohort study, all consecutive SAH patients that were treated at our institution between January 2003 and June 2016 and were eligible were included. Patients were considered to be eligible when they: (1) were above 18 years of age and (2) had an available pre-treatment computer tomography (CT) scan < 48 h after the aneurysm rupture, allowing the measurement of several ventricular indices. Exclusion criteria were: (1) no SAH treatment received by the patient; (2) no available CT scan or a scan > 48 h after ictus. Our study is imbedded in the clinical trial registered at the German trial registry (DRKS, Unique identifier: RKS00008749). The study was approved by the local ethics committee (Ethik-Kommission, Medizinische Fakultät der Universität Duisburg-Essen, Registration number: 15-6331-BO).

### 2.2. SAH Management

All patients with SAH who presented at our facility were admitted to the neurosurgical intensive care unit. Rupture of an intracranial aneurysm was confirmed in digital subtraction angiography (DSA), and aneurysms were usually treated within 24 h either by microsurgical clipping or endovascular coiling. In our previous works, our protocol for post-interventional management at the intensive care unit was described [10,11,12]. In summary, the conservative treatment is composed of maintenance of the mean arterial pressure > 70 mmHg, oral nimodipine for the first 21 days after ictus and consistent normovolemia. External ventricular drainage (EVD) was placed in case of acute hydrocephalus. In case of pathologically increased intracranial pressure (ICP) > 20 mmHg, conservative treatment was initiated, consisting of deep sedation, forced cerebrospinal fluid (CSF) drainage and osmotherapy. As previously described [12], SAH patients with: (a) clinical, intraoperative and/or radiographic signs of intracranial hypertension at admission, and (b) ICP increase in the course of SAH refractory-to-maximal conservative treatment underwent primary and secondary decompressive craniectomy (DC), respectively. Vasospasm management consisted of a daily neurological examination and transcranial Doppler (TCD) sonography, with the endovascular treatment of SAH patients with symptomatic cerebral vasospasm [11]. Laboratory tests of blood and CSF were performed routinely at admission and three times per week, with additional tests performed if they were clinically indicated. CT scans were conducted in case of any neurological deterioration after surgical procedures and around the placement of a permanent CSF diversion. 

### 2.3. Data Management

All available pre-treatment CT scans after admission were reviewed by the first author (M.S.), who was blinded at this time to clinical information. Measurements of the ventricular indices were conducted as previously described. This concerned the bifrontal, bicaudate, ventricular and third ventricle ratios, as well as Evans’ and Huckman’s indices [3,4,13,14,15,16]. A graphical overview of the exact measurements was presented in our previous work (see Appendix A Appendix A). 

Furthermore, various demographic (age and sex), clinical (World Federation of Neurosurgical Societies (WFNS) scale [17] at admission and treatment modality), laboratory and radiographic (presence of intraventricular hemorrhage, Fisher’s scale [18] and Subarachnoid Early Brain Edema Score (SEBES) [19] at admission) data of patient and SAH characteristics, complications during SAH and outcomes were collected from the institutional aneurysm database. For the assessment of the inflammatory status during SAH, the following laboratory parameters were included (admission and 14 days mean values): white blood cells (leukocytes) and c-reactive protein (CRP) in blood and interleukin-6 (IL-6) in CSF. The clinical/adverse events also recorded for further analysis were: aneurysm rebleeding before treatment, early angiographic vasospasm (documented within 72 h after ictus), occurrence and duration of cerebral vasospasm (s) in TCD (mean flow velocities > 120 cm/s), development of delayed ischemic neurological deficit (DIND), endovascular treatment for vasospasm, epileptic seizures, new cerebral infarction (s) in follow-up CT scans, occurrence of a fever (body temperature > 38.0° at admission and during SAH), duration of ICP increase, need for DC, duration of mechanical ventilation (including the duration of sedation for mechanical ventilation), development of meningitis, systemic infections, acute coronary syndrome and acute renal failure during SAH. For the assessment of the functional outcome of SAH, the rates of in-hospital mortality and unfavorable outcome at 6 months after SAH (defined as modified Rankin Scale > 3 [20]) were included in the analysis. 

### 2.4. Study Endpoints and Statistical Analyses

We aimed to predict the correlation between ventricular morphology and the occurrence of complications in the course of SAH and its final outcome by using measurements of the above-mentioned indices. Our main study endpoints were cerebral infarction, in-hospital mortality and an unfavorable outcome at 6 months. Secondary endpoints included the associations between ventricular indices and the occurrence of the above-mentioned adverse events in the clinical course. The predictive value of ventricular measurements of chronic hydrocephalus and shunt dependency after SAH was analyzed elsewhere.

First, the ventricular indices were evaluated as continuous variables in an explorative manner using the Pearson correlation and the Spearman’s rank tests, as appropriate. For further univariate and multivariate analyses, all ventricular indices were dichotomized at the cut-offs defined using the area under the curve (AUC) on the receiver operating characteristic (ROC) analyses for prediction of the primary study endpoints. Thereafter, univariate analyses for correlations between the dichotomized ventricular indices and primary and secondary study endpoints were performed. For categorical variables, the Chi-square test (χ^2^ test) or Fisher’s exact test, as appropriate, was used. For continuous variables, Student’s *t*-test (for normally distributed data) or Mann–Whitney U test (for non-normally distributed data) was applied. Then, multivariate analysis was performed on the significant findings in order to confirm the independent association. Four baseline parameters associated with ventricular indices—age, SEBES, WFNS and Fisher’s scores at admission—were included as obligatory components of the multivariate models. Furthermore, the significant ventricular indices were included in a stepwise backward regression model. The associations between ventricular morphology and the burden of ICP increase after SAH were additionally analyzed using Kaplan–Meier survival plots, and the construction of a novel prediction score was achieved using the results of the multivariate analysis. Data analysis was performed using SPSS statistical software (version 25, SPSS Inc., IBM Company, Chicago, IL, USA). Correlations with a *p*-value of <0.05 were considered to be statistically significant.

## 3. Results

A total of 745 SAH patients in the above-mentioned study period were eligible for our study, and thus, were included in the final cohort. We provide an overview of major demographic, clinical and radiographic characteristics in Table 1.

Initial explorative analyses revealed an association between the ventricular indices as continuous variables, and several recorded parameters (see Appendix A Appendix A) showed significant results for patients’ demographic characteristics, initial SAH severity and different secondary complications, as well as the primary study endpoints. On the whole, the larger the ventricles at admission were, the poorer the patient’s functional outcome was. SAH patients with larger ventricles were more likely to be of older age and present with increased SAH severity (higher WFNS, SEBES and Fisher’s grades), but they were at a lower risk of ICP- and vasospasm-related complications during SAH.

For further analyses, clinically relevant cut-offs for the ventricular indices were defined using ROC analyses: bifrontal ratio ≥ 0.337; bicaudate ratio ≥ 0.175; Evans’ index ≥ 0.281; ventricular ratio ≥ 0.559; Huckman’s index ≥ 5.48; third ventricle ratio ≥ 0.053 (see Appendix A Appendix A). Univariate analyses of the dichotomized ventricular indices and the primary/secondary endpoint events are provided in Appendix A Appendix A.

Finally, all the primary study endpoints and significant adverse events from the univariate analyses were tested by multivariate analysis, which was adjusted for patients’ age and initial SAH severity (Table 2). These analyses confirmed an independent association between larger ventricles with a risk of cerebral infarction (ventricular ratio ≥ 0.559: adjusted odds ratio (aOR) = 1.54, *p* = 0.017), in-hospital mortality (bicaudate ratio ≥ 0.175; aOR = 1.61; *p* = 0.025) and unfavorable outcome at 6 months (Evans’ index ≥ 0.281: aOR = 1.67, *p* = 0.017; ventricular ratio ≥0.559: aOR = 1.73, *p* = 0.013). 

Regarding the secondary endpoints of the study, the multivariate analysis also showed an independent association between larger ventricles and a lower ICP burden after SAH: lower risk of ICP increase > 20 mmHg (third ventricle ratio ≥ 0.053: aOR = 0.65, *p* = 0.017) and the need for DC (ventricular ratio ≥ 0.559: aOR = 0.43, *p* = 0.001). Figure 1 shows the duration of ICP increase in different SAH subpopulations depending on the values of ventricular indices. In the Kaplan–Meier survival analysis, the ventricular ratio ≥ 0.559 proved to be the best fit for prediction of the need for and timing of DC after SAH (Figure 2). Based on the results of the multivariate analysis for the prediction of DC, we constructed a risk score based on significant parameters and appropriate aOR values. Accordingly, the novel risk score for DC prediction (0–5 points) included the following components: WFNS grade 4–5 (1 point), Fisher’s grade 3–4 (2 points), SEBES grade 3–4 (1 point) and the ventricular ratio < 0.559 (1 point, see Table 3). The constructed risk score showed a good diagnostic accuracy for the prediction of DC (AUC = 0.754). With increasing points on this risk score, the patients’ need for DC increased in the analyzed cohort (see Figure 3).

From the remaining multivariate analyses, the ventricular indices showed no link between the risk of aneurysm rebleeding, cerebral vasospasm on TCD or DSA and the occurrence of systemic complications during SAH, such as systemic infections, acute coronary syndrome and acute renal failure. At the same time, larger ventricles were significantly and independently associated with several parameters of highly inflammatory responses at the beginning and during the course of SAH, including CRP in the blood (Evans’ index ≥ 0.281: aOR = 1.55, *p* = 0.011; ventricular ratio ≥ 0.559: aOR = 1.49, *p* = 0.031) and IL-6 in CSF (third ventricle ratio ≥ 0.053: aOR = 1.69, *p* = 0.047), as well the duration of a fever (Evans’ index ≥ 0.281: aOR = 1.55, *p* = 0.012, see also Figure 4). Of note, the association between the third ventricle ratio and IL6 course in CSF was present, independent of the occurrence of bacterial meningitis during SAH.

## 4. Discussion

To our knowledge, this is the first study to evaluate the value of ventricular indices for the prediction of different complications and the outcome of SAH. We showed that the initial ventricular morphology after the bleeding event is strongly associated with the functional outcome of SAH. At the same time, patients with larger ventricles showed fewer complications related to the pathological ICP increase and no link with cerebral vasospasm, but a remarkable inflammatory response at the beginning and during the course of SAH.

Aneurysm rupture leads to a cascade of pathophysiologic processes at the molecular and cellular levels, such as cortical spreading depolarization, microcirculatory failure, cytotoxic and vasogenic edema, oxidative stress, microthrombosis and apoptosis, which later transform into clinically evident complications, such as ICP increase and the development of delayed cerebral ischemia (DCI) [21]. The occurrence and severity of these outcome-relevant pathophysiologic processes depends, at least partially, on the extent of the local and systemic inflammatory responses to the aneurysmal bleeding and the subsequent early brain injury [22]. Neuroinflammation within the brain is driven by inflammatory cells, both those that are locally present and circulating peripherally, and mediated by reactive oxygen species, cytokines, chemokines and other messenger molecules [23,24,25].

The first studies analyzing inflammation after SAH have attempted to identify a link between inflammation and the risk of cerebral vasospasm [26,27]. Recent studies also identified a higher risk of DCI with increasing inflammatory response after SAH [28,29]. However, this association is probably due to the involvement of inflammatory mediators in the coagulation cascades, resulting in microthrombosis and microcirculatory disturbances, and accordingly, a higher risk of DCI, rather being than related to the vasospasm of large intracranial vessels [30]. Fittingly, several studies have shown a robust association between the level of inflammatory markers in peripheral blood and the occurrence of cerebral infarcts and a poor outcome after SAH [31]. Therefore, the inhibition of inflammation after SAH with medical agents might present a therapeutic target for the prevention of a secondary brain injury after SAH. There are data from small cohort studies that show the positive clinical effect of glucocorticoid treatment in SAH patients [32,33,34]. This effect might show up in the early phase of SAH by the suppression of microenvironmental and systemic cytokine releases involved in different pathways of an early brain injury [35,36]. Moreover, the increased downregulation of inflammatory cytokines by continuous dexamethasone administration can also contribute to the prevention of secondary SAH complications such as DCI. To assess the above-mentioned results and other potential mechanisms of the glucocorticoids effect, the organizers of a multi-center prospective trial analyzing the effect of treatment with dexamethasone on the SAH outcome are currently recruiting patients in Germany [29].

In the context of the above-described evidence on the impact of inflammation on SAH, the early identification of individuals with a pronounced inflammatory response to SAH is of paramount clinical importance. So, the timely and correct selection of SAH individuals who would benefit from an anti-inflammatory treatment would help to improve the outcome of affected patients, without the overtreatment of cases not showing more extensive inflammation. This circumstance underlines the relevance of our study results. We showed the association between larger ventricles at admission and a higher expression of inflammatory parameters in the blood and CSF, as well as longer duration of a fever during SAH. The exact pathophysiologic background of the link between initial ventricle enlargement and inflammatory response after SAH is unclear. Previous animal studies reported that the association between posthemorrhagic hydrocephalus and elevated inflammation might be conditioned, at least in part, by elevated levels of activity of choroid plexus membrane transporters involved in CSF secretion [37]. According to our results, the ventricle morphology at admission seems to be an early and reliable radiographic marker of the severity of SAH-induced inflammation. The fact that there was no association between the ventricle measurements and systemic infections might indicate the specificity of ventricular indices for inflammatory processes within the injured brain. The absence of a link between larger ventricles and a high risk of ICP- and vasospasm-related complications after SAH also accounts for the decisive role of neuroinflammation as the mediator of the observed association between larger ventricles and the poor outcome of SAH in our cohort. Further clinical and experimental studies to confirm our findings are needed.

So far, the role of ventricular morphology for SAH patients has been addressed only with regard to the prediction of shunt dependency in two small SAH cohorts [38,39]. In our study, along with the independent correlation found between the poor functional outcome of SAH and larger ventricles at admission, we also showed an inverse association between ventricle size and the risk of ICP-related complications. In particular, the larger the ventricles are after aneurysm bleeding on admission CT scans, the lower the risk of ICP increase requiring conservative and surgical treatments is. The duration of pathologic ICP increase was also shorter in SAH individuals with larger ventricles. This inverse association of ventricle morphology with documented ICP increases might be explained by the more pronounced ICP-reducing effect of EVD in these individuals, helping to control the ICP without (or at least with less) documentation of a pathologic ICP increase. As result, SAH patients with initially enlarged ventricles required less frequent additional medical or surgical interventions to reduce the ICP increase.

The observed association between ventricle size and the burden of ICP after SAH is also of clinical relevance. In particular, we constructed a risk score for the need of DC after SAH using independent predictors from the multivariate analysis, including ventricle morphology. The presented novel score based on early and easily assessable clinical and radiographic parameters showed a high diagnostic accuracy for the prediction of DC in SAH patients. The timely selection of SAH individuals needing DC might be helpful in their outcome improvement, as it has been shown in a recent study [12]. Prior to clinical application, an external validation of the presented risk score is necessary. 

In summary, provided that we obtain confirmation of our findings on the value of ventricle morphology as an early reliable radiographic marker of the post-SAH inflammatory response and the need for DC in future experimental and clinical studies, the measurement of ventricular indices at admission might be included in decision algorithms for SAH patients. In particular, individuals with severe SAH (WFNS grade 4–5 and Fisher’s grade 3–4) with ventricular ratio ≥ 0.559 might benefit from anti-inflammatory treatment (e.g., with glucocorticoids), whereas SAH individuals with a narrow ventricle system (ventricular ratio < 0.559) might be more eligible for early DC.

### Limitations

The retrospective and monocentric design of our study are its biggest limitations, resulting in a number of potential research biases that cannot be fully ruled out. Therefore, no judgment on a true causal relationship between the significant study results can be made; thus, different hypotheses may be proposed. Although they are currently of rather speculative nature, these hypotheses might become future targets for clinical and experimental research. Nevertheless, our study is based on a large representative SAH cohort. The analyses included different statistical assessments to address the potential confounding effect of other parameters. The findings of our study require validation in future clinical and experimental studies.

## 5. Conclusions

According to our study results, SAH patients with larger ventricles are at a higher risk of cerebral infarctions, in-hospital mortality and an unfavorable outcome at 6 months after SAH. The absence of a correlation with higher ICP or vasospasm risk values indicates that the observed outcome effect might be related to a more extensive inflammatory response documented in SAH individuals with larger ventricles. Therefore, ventricular indices might be a valuable radiographic marker of post-SAH neuroinflammation, helping to identify the individuals who are more likely to benefit from therapeutic anti-inflammatory measures. The diagnostic value of ventricle morphology for DC prediction might aid the timely selection of SAH patients requiring DC.

## Figures and Tables

**Figure 1 jcm-12-02585-f001:**
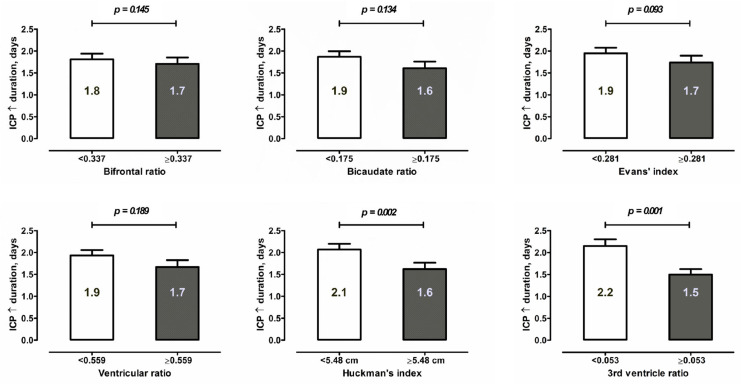
Mean duration of ICP increase after SAH (in days) in different subgroups depending on the values of the ventricular indices. The larger the ventricles are, the shorter the duration of ICP increase > 20 mmHg requiring medical or surgical treatment is.

**Figure 2 jcm-12-02585-f002:**
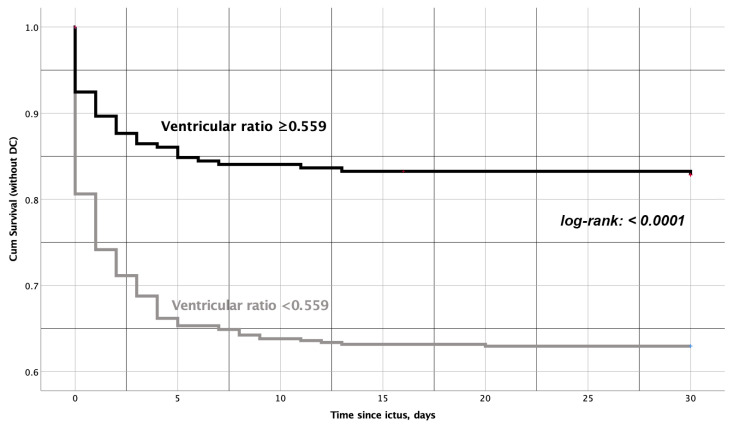
Kaplan–Meier survival plot showing the diagnostic value of the ventricular ratio (with cut-off of 0.559) for predicting the need for DC and its timing. The probability to survive SAH without DC was significantly higher in patients with larger ventricles. SAH patients with smaller ventricles underwent DC more frequently.

**Figure 3 jcm-12-02585-f003:**
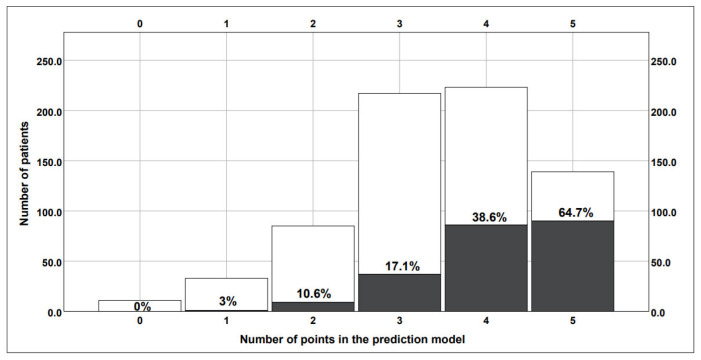
Risk prediction model for the need of DC in SAH patients based on four independent predictors from the multivariate analysis (see Appendix A Appendix A): SEBES grade 3–4, ventricular ratio < 0.559, WFNS grade 4–5 and Fisher’s grade 3–4. With every point increase in this score, the probability for the need for DC increases. Grey = percentage of patients in need of DC out of all patients with the same score (white ones).

**Figure 4 jcm-12-02585-f004:**
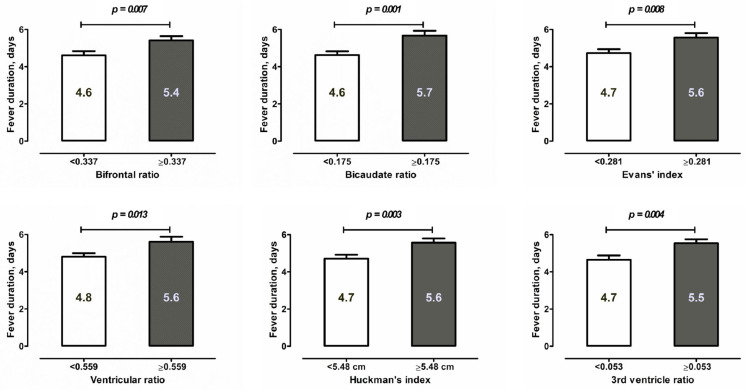
Mean duration of fever after SAH (in days) in different subgroups depending on the values of the ventricular indices. Larger ventricles are significantly and independently associated with a longer duration of fever.

**Table 1 jcm-12-02585-t001:** Major demographic, clinical and radiographic characteristics of the final cohort.

Parameter	Number of Cases (% *) or Mean (±SD)
Ventricular measurements	
Bifrontal ratio: A/a	0.339 (±0.057)
Bicaudate ratio: B/b	0.166 (±0.081)
Ventricular ratio: B/A	0.517 (±0.214)
Third ventricle ratio: C/c	0.062 (±0.046)
Evans’ index: A/D	0.275 (±0.046)
Huckman’s index: A + B (cm)	5.419 (±1.244)
Demographic characteristics	
Age (years)	54.7 (±13.9)
Sex (female)	497 (66.7%)
SAH characteristics	
WFNS (grade 4–5)	360 (48.3%)
SEBES (grade 3–4)	386 (52.1%)
Fisher’s scale (grade 3–4)	638 (90.1%)
Clipping	302 (40.5%)
Presence of IVH	390 (52.4%)
Acute hydrocephalus	570 (76.5%)
Neurologic complications during SAH	
Aneurysm rebleeding	49 (6.6%)
DIND	193 (31.3%)
Pathologic ICP increase	368 (49.9%)
Decompressive craniectomy	228 (30.6%)
TCD vasospasm	340 (53.0%)
Early angiographic vasospasms (within 72 h)	57 (9.7%)
Endovascular treatment of vasospasm	171 (23.0%)
Epileptic seizures	68 (9.1%)
Systemic complications during SAH	
Systemic infections	310 (46.3%)
Acute coronary syndrome	20 (3.1%)
Acute kidney failure	9 (1.4%)
Primary study endpoints	
Cerebral infarction (s)	380 (51.4%)
In-hospital mortality	159 (21.3%)
Unfavorable outcome at 6 months (mRS > 3)	296 (42.9%)

Abbreviations: A = maximum width between the two frontal horns; a = internal width of the vault at level of A, B = minimum width of the ventricles between caudate nuclei; b = internal width of the vault at level of B, C = greatest width of the third ventricle; c = internal width of the vault at level of C, D = maximum internal width of the vault. SD = standard deviation; IVH = intraventricular hemorrhage; DIND = delayed ischemic neurological deficit; TCD = transcranial Doppler sonography; SEBES = subarachnoid hemorrhage early brain edema score; WFNS = world federation of neurosurgical societies; DCI = delayed cerebral ischemia; mRS = modified Rankin scale. * Percentages were calculated using the cases with known values.

**Table 2 jcm-12-02585-t002:** Multivariate analysis for the predictors of the primary study endpoints (cerebral infarction, in-hospital mortality and unfavorable outcome at 6 months after SAH). The model obligatory includes the major baseline confounders (age, WFNS, Fisher’s and SEBES grades) displayed as ventricular indices in the backward regression model. Cut-offs of ventricular indices, determined by the AUC according to ROC analyses, were used. For all analyses, only the last steps were included.

Ventricular Ratio/Index	aOR (95% CI)	*p*-Value
Cerebral infarction		
Age >55 years	1.39 (0.97–1.99)	0.073
WFNS grade 4–5	**2.57 (1.82–3.62)**	**<0.0001**
SEBES 3–4	**1.61 (1.13–2.28)**	**0.008**
Fisher’s grade 3–4	1.50 (0.84–2.69)	0.174
Ventricular ratio ≥0.559	**1.54 (1.07–2.19)**	**0.017**
In-hospital mortality		
Age >55 years	**1.81 (1.17–2.79)**	**0.007**
WFNS grade 4–5	**3.57 (2.28–5.58)**	**<0.0001**
SEBES 3–4	1.49 (0.96–2.31)	0.074
Fisher’s grade 3–4	4.12 (0.96–17.67)	0.057
Bicaudate ratio ≥0.175	**1.61 (1.06–2.44)**	**0.025**
Unfavorable outcome at 6 months after SAH *		
Age >55 years	**2.75 (1.78–4.23)**	**<0.0001**
WFNS grade 4–5	**7.03 (4.69–10.56)**	**<0.0001**
SEBES 3–4	**1.73 (1.12–2.69)**	**0.013**
Fisher’s grade 3–4	**4.68 (1.57–13.97)**	**0.006**
Evans’ index ≥0.281	**1.67 (1.10–2.53)**	**0.017**
Ventricular ratio ≥0.559	**1.73 (1.12–2.65)**	**0.013**

Abbreviations: aOR = adjusted odds ratio; CI = confidence interval; AUC = area under the curve; ROC = receiver operating characteristic. SEBES = subarachnoid hemorrhage early brain edema score; WFNS = world federation of neurosurgical societies. * Defined as modified Rankin scale >3. Significant values are in bold.

**Table 3 jcm-12-02585-t003:** Components and weights of parameters in the prediction model for decompressive craniectomy.

Parameter	Score Weight
WFNS grade 4–5	1
SEBES 3–4	1
Fisher’s grade 3–4	2
Ventricular ratio <0.559	1

Abbreviations: WFNS = world federation of neurosurgical societies; SEBES = subarachnoid early brain edema score.

## Data Availability

Data are available upon reasonable request by contacting the first author.

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
