# Peer review of "Morphometric Study of the Initial Ventricular Indices to Predict the Complications and Outcome of Aneurysmal Subarachnoid Hemorrhage"

_jcm, 2023, doi:10.3390/jcm12072585_

Round 1

Reviewer 1 Report

The authors performed a retrospective, monocentric analysis of 745 patients with SAH needing neurosurgery intervention and evaluated the association between ventricular indices with clinical course, occurrence of complications and outcome of SAH.They conclude that patients with larger ventricels are at higher risk of cerebral infarctions, in-hospital mortality and unfavorable outcome at 6 month, but have a lower risk of intracranial pressure increase and absence of an association with vasospasm. The theory is that this is related to a higher extent of inflammatory response in patients with larger ventricels.

All in all, the results presented here are interesting. The topic is timely, while the study contributes kind of novel information regarding the identification of individuals who are more likely to benefit from therapeutic anti-inflammatory measures and the prediction of needing decompressive craniectomy.

The results presented in this study seem robust.The length of the manuscript is a little bit too long.

I have just few comments/questions:

1. The Hunt and Hess scale is more common in the clinical classification of SAH, even if the interobserver variability is high. Please comment.

2. To me as a non-statistician the applied methodology appears state of the art and meticulously employed. Just one comment: . If multiple testing is applied, the effects of the p values will probably be reduced through multiple testing corrections.Please comment.

3. Is there a correlation between the prognosis due to the different localisations of aneurysms in your study?

4. . Increased IL-6 and CRP levels after SAH may be a consequence of vasospasm. Conversely, Dexamethasone administration decreased the serum IL-6 and CRP levels in the SAH, eventually attenuating cerebral vasospasm and improving neurological deficit outcomes (see Rothoerl et al, JNeurosurg Anesthesiol (2006) 18:68–72.) So the Dexamethasone auxiliary role in SAH microenvironment cytokines and etiological factor regulation may ameliorate early-phase SAH symptoms… Can you comment on that ?

5. Would add Jin et al, Frontiers in Immunology 2022

Author Response

Please see attached document below.

Reviewer 2 Report

The article is relevant primarily for clinicians. For the first time, a detailed analysis of the influence of initial ventricular indices in SAH on the prognosis of the development of cerebral infarctions, in-hospital mortality and unfavorable outcome at 6 months are given. I see further improvement of this article in the creation of an algorithm of actions of a clinician when measuring initial ventricular indices. It would be nice to draw a diagram which would indicate the sequence of necessary procedures after detecting certain combinations of ventricular indices.

Author Response

Please see attached document below.

Reviewer 3 Report

Ventricular indices were independently associated with the parameters of inflammatory response after SAH (C-reactive protein in serum, interleukin-6 in cerebrospinal fluid and fever). There were independent correlations between larger ventricles and all primary endpoints.
We know that if a person's blood vessels are in poor condition or as they age, they can develop brain atrophy, which can affect the indicators of the ventricles,how the authors remove these effects
How ventricular indices were associated with all primary endpoints need more, deeper discussions

Author Response

Please see attached document below.
